# Enhancing the Resilience of Agroecosystems Through Improved Rhizosphere Processes: A Strategic Review

**DOI:** 10.3390/ijms26010109

**Published:** 2024-12-26

**Authors:** Waleed Asghar, Kelly D. Craven, Jacob R. Swenson, Ryota Kataoka, Ahmad Mahmood, Júlia Gomes Farias

**Affiliations:** 1Department of Biochemistry and Molecular Biology, Oklahoma State University, Stillwater, OK 74078, USA; kelly.craven@okstate.edu (K.D.C.); jswenso@okstate.edu (J.R.S.); 2Department of Environmental Sciences, Faculty of Life & Environmental Sciences, University of Yamanashi, Yamanashi 400-0016, Japan; rkataoka@yamanashi.ac.jp; 3Departments of Climate Change and Soil and Environmental Sciences, Muhammad Nawaz Shareef-University of Agriculture, Multan 60000, Pakistan; ahmad.mahmood@mnsuam.edu.pk; 4USDA-ARS, US Arid Land Agricultural Research Center, 21881 North Cardon Lane Maricopa, Maricopa, AZ 85138, USA; julia.stiles@usda.gov

**Keywords:** sustainable agroecosystems, rhizosphere process, soil microbes, green manure, intercropping

## Abstract

As farming practices evolve and climate conditions shift, achieving sustainable food production for a growing global population requires innovative strategies to optimize environmentally friendly practices and minimize ecological impacts. Agroecosystems, which integrate agricultural practices with the surrounding environment, play a vital role in maintaining ecological balance and ensuring food security. Rhizosphere management has emerged as a pivotal approach to enhancing crop yields, reducing reliance on synthetic fertilizers, and supporting sustainable agriculture. The rhizosphere, a dynamic zone surrounding plant roots, hosts intense microbial activity fueled by root exudates. These exudates, along with practices such as green manure application and intercropping, significantly influence the soil’s microbial community structure. Beneficial plant-associated microbes, including *Trichoderma* spp., *Penicillium* spp., *Aspergillus* spp., and *Bacillus* spp., play a crucial role in improving nutrient cycling and promoting plant health, yet their interactions within the rhizosphere remain inadequately understood. This review explores how integrating beneficial microbes, green manures, and intercropping enhances rhizosphere processes to rebuild microbial communities, sequester carbon, and reduce greenhouse gas emissions. These practices not only contribute to maintaining soil health but also foster positive plant–microbe–rhizosphere interactions that benefit entire ecosystems. By implementing such strategies alongside sound policy measures, sustainable cropping systems can be developed to address predicted climate challenges. Strengthening agroecosystem resilience through improved rhizosphere processes is essential for ensuring food security and environmental sustainability in the future. In conclusion, using these rhizosphere-driven processes, we could develop more sustainable and resilient agricultural systems that ensure food security and environmental preservation amidst changing climate situations.

## 1. Introduction

The swift increase in the global population has markedly heightened the need for energy, food, and other essential goods, leading to the expansion of agricultural land use. Agroecosystems are intricate, evolving environments in which agricultural activities engage with natural ecosystems, affecting soil health, biodiversity, and ecosystem services. Recent studies highlight the significance of agroecosystems in enhancing resistance to climate change, boosting soil health, and fostering biodiversity [1,2]. For example, soil is a complex ecosystem and is one of our most important resources due to its many vital functions, such as providing food, fuel, and fiber; recycling nutrients that turn into beneficial nutrients for crop and plant necessities; organic matter decomposition, including plants and animal residues; regulation of water supply and quality; and provision of soil microorganisms, which are important for soil health and play vital roles in agroecosystems [3]. However, poor farming techniques, urbanization, and inadequate soil management practices have had negative impacts on agricultural soil, such as reduced soil fertility, the loss of beneficial soil microbiota, and the migration of soil organisms [4]. The top layer of soil surrounding plant roots, known as the rhizosphere, is home to a diverse range of microbes that play vital roles in promoting plant growth, organic matter degradation, and maintaining soil sustainability. However, in the lack of proper soil management, coupled with inappropriate agricultural practices, the quality of rhizosphere soil deteriorates, and its microbial communities weaken [5,6].

The rhizosphere serves as a microbial hotspot owing to the exudates emitted by plant roots, which supply vital nutrients for microbial activities [7]. Dynamics, for example, root zones, plant growth phases, plant species, and environmental conditions, change the diversity and composition of soil microbial communities (e.g., bacterial and fungi) in the rhizosphere [8]. Multiple species can compose microbial groups, demonstrating comparable mechanisms of action yet differing in their tolerance to plant cultivars and various environmental stresses. The microorganisms found in the rhizosphere include fungi, bacteria, and nematodes [9]. Large numbers of this community have an impartial effect on soil and plants, but some microbes are directly or indirectly connected with the plant–soil systems and provide nutrients from the rhizosphere to the plant. Additionally, rhizosphere microbiota contains microorganisms with both positive and negative effects on plant growth and soil health. Harmful microbes include pathogenic fungi, nematodes, and bacteria, while beneficial microbes, such as PGPM, nitrogen-fixing bacteria, and mycorrhizal fungi, contribute to plant nutrition and soil health [9,10]. However, beneficial microbes provide essential plant nutrients and contribute to lowering greenhouse gas emissions, improving nutrient availability, suppressing soil-borne plant diseases, and balancing nutrient cycling [11].

Conversely, many authors have pointed out that the use of various agricultural techniques, overuse of mineral fertilizers, agrochemicals, and farming practices could reduce soil fertility and disturb the rhizosphere microbiota, resulting in decreased soil health and crop production [12,13]. In general, overuse of agrochemicals (i.e., fungicides, herbicides, nematicides, and insecticides) is being used non-judiciously, which has a harmful effect on soil-positive microbiota in the soil as well as the overall health and quality of the soil [14,15]. This impact is mostly related to changes in microbial parameters such as enzymatic activity and biomass, which are the most important features of soil health and are important tools in monitoring soil quality. Eighty to ninety percent of soil processes are controlled by soil microorganisms and are strongly linked with plant and soil health, in addition to the role of microorganisms in the suppression of soil-borne diseases, nutrient cycling, soil resilience, and maintenance of soil sustainability [16,17]. Nevertheless, several authors reported that green manure is an alternate way to sustain agriculture and maintain soil health. The integration of the entire plant, containing more nutrients than crop straw, into the soil enhances soil fertility by adding more organic matter. Green manure improves crop yield and growth, reduces pests and insects, suppresses plant diseases, and increases soil enzyme activities, which regulate nutrient fluxes and stimulate plant growth [18,19,20]. Therefore, a sustainable solution must be sought by utilizing all available resources to modulate the rhizospheric process for sustainably enhancing agroecosystem productivity.

It is important for long-term sustainability in agricultural systems to keep agroecosystems stable and improve rhizosphere processes that make soil more resilient. This includes keeping and expanding the diversity of microorganisms in the rhizosphere. The rhizosphere is crucial for facilitating nutrient cycling, enhancing soil structure, and promoting plant health. The primary goal of this review is to investigate several methods for improving agricultural ecosystem resilience through rhizosphere activity optimization. We can implement these practices by utilizing beneficial microbes, green manures, and intercropping. These strategies jointly enhance synergistic communications between soil microorganisms and plant roots, creating an environment favorable for soil health and plant growth. The findings of this review improve comprehension of sustainable agriculture techniques, highlighting the importance of natural and organic resources in boosting crop production while preserving ecological balance.

## 2. The Rhizosphere: Its Mechanism and Roles

The rhizosphere refers to the area surrounding the root zone [18,19]. This zone is considered essential for biological activities because many exudates produced by plant roots provide nutrients for soil microorganisms. Thus, this region is vital for disease resistance, nutrient cycling, and plant growth [20]. The rhizosphere zone has a central role in the food chain of soil microorganisms. The microbial food chain mainly includes dead plant matter and living plant roots, which are the primary source of nutrients [21], and carbon from living plant tissues and root exudates. These are crucial for plant health and growth as they support disease suppression, protection, and nutrient uptake [22]. The activity of beneficial microorganisms in the rhizosphere not only provides essential nutrients to plants but also influences the quality and quantity of root exudates [22,23]. Furthermore, various metabolites exuded by plant roots into the soil primarily shape and attract the microbial community composition in the rhizosphere, facilitating interactions with both biotic and abiotic factors. Plants frequently adjust the diversity of these microbes according to the benefits they provide for growth and health [24]. In the rhizosphere, plants engage in a wide range of interactions with soil-dwelling organisms, including exploitation, competition, neutrality, mutualism, and commensalism. These diverse interactions are essential for plants to adapt to varying environmental conditions, which is crucial for their survival and agricultural productivity. In the rhizosphere, plants host a diverse collection of microbes, and they can inhabit the inner tissues (endosphere), and surface tissues (phylloplane and rhizoplane). Collectively, these microorganisms are referred to as the plant microbiome [25,26]. The plant and soil microbiome is critical for stimulating plant development, improving soil quality and fertility, and supporting agricultural sustainability by strengthening resilience to environmental stressors, nutrient cycling, and disease-fighting [27,28]. The interplay between roots and their surrounding soil constitutes one of the most prolific and diverse ecosystems in nature. The rhizosphere, the tiny and thin soil layer around the roots, has a vast population of microbes fueled by carbon-rich compounds exuded during photosynthesis and provides essential nourishment for the recruited microorganisms [29]. Overall, the rhizosphere is a kind of microenvironment wherein plant root development and nutrient uptake, soil properties, and microbe activities interact in a corresponding manner (Figure 1).

### 2.1. Rhizosphere Processes: Nutrient Retention in Agroecosystems

The rhizosphere hosts diverse microbial groups that play critical roles in nutrient cycling, promoting plant growth, and protecting plants from pathogens, as well as from biotic and abiotic stresses [21]. Changes in rhizosphere processes directly contribute to nutrient retention in agroecosystems. Through plant–soil interactions, plants and their residues provide organic carbon (C) to the soil. The rate of soil carbon input depends primarily on the rate of plant growth, which is driven by photosynthesis. Root-derived carbon input, in particular, is a key regulatory factor in the plant–soil interaction within the rhizosphere, as it has a more direct impact on soil processes than carbon input from shoots [22]. This root-derived carbon plays a significant role in soil carbon sequestration. The amount of carbon that can be stored as soil organic matter (SOM) depends on the balance between carbon input and the rate at which plant materials decompose. Decomposition is the process by which microorganisms break down organic molecules into their inorganic components, such as nutrients and CO_2_. Carbon dioxide, produced during microbial respiration, is released from the soil back into the atmosphere. However, some decomposition products are protected from further breakdown, playing a critical role in forming stable SOM, which is essential for long-term carbon sequestration in agroecosystems [22]. As a result, the rhizosphere process is particularly crucial for nutrient cycling, stabilizing SOM, making nutrients available for plant growth, and ultimately handling soil microorganisms. On the other hand, maintaining and increasing soil and its process through utilizing carbonized materials such as “Biochar” have long-term stability [23]. Biochars can reduce the emission of N_2_O and CH_4_ from the soil [24]. Biochar comprises a stable C compound as a derivative of gasification and pyrolysis and is available either as a pelletized or powder form. Char is ubiquitous in natural soils, containing as much as 35% of total organic C (TOC) in fire-impacted soils. Laird [25] reported that the application of biochar into the soil was found to enhance soil fertility and significantly increase soil organic carbon (SOC) and nutrients, which leads to the improvement in soil health. SOC consists of a complex mix of partially decomposed substances, including polysaccharides, lignin, aliphatic biopolymers, tannins, lipids, proteins, and amino sugars derived from plant litter, as well as microbial and faunal biomass [26]. However, SOC has been classified into slow, active, and inert pools, depending upon the decomposition rate in soil [27]. Furthermore, biochar largely supports the SOC pool for stabilization, while green manure incorporation adds to the more bio-accessible fraction of the soil C reserve. This is because biochar decomposes typically slower than uncharred biomass, e.g., green manure incorporation and fresh crop residues [28]. However, green manure or fresh crop residues in agricultural fields, on the other hand, can be used as a SOC storage/sequestration strategy by supporting cover crops; conservation tillage provides benefits including nutrient cycling, control of surface runoff of water, wind erosion, and crop production [29]. Yazdanpanah [30] reported that applying green manure or plat residues improved the stability and total porosity of soil aggregates and increased the soil C pool, but this depended on the type and application rate of soil amendments. However, Kenney et al. [31] reported that the removal of crop residues (e.g., corn stover and *Zea mays* L.) decreased the soil C pool. Therefore, the following strategy should be considered to achieve higher soil C sequestration via soil amendments produced from green manure/crop residues or biochar. The indirect effects of using these amendments should be considered in the context of production/control of soil GHG, environmental health, and economic cost benefits of their use. As per our knowledge, the combined effect of biochar and crop residues has rarely been investigated except in one recent study by Nguyen et al. [32], and it should be considered in the near future. Overall, biochar seems to have a greater positive impact on total soil SC, while the incorporation of crop residues/green manures can increase the microbial biomass C of soil, enhance the rhizosphere process, and reduce carbon dioxide emission.

### 2.2. Rhizosphere Engineering: Turning to Sustainably Increases the Agroecosystem and Its Productivity

The rhizosphere is a hotspot of microbes because many exudates are produced by plant roots, which are the primary source of nutrients for essential microbial processes and sustain the agroecosystems and their functions. The rhizosphere contains diverse microbial groups that perform various functions and affect nutrient cycling and plant growth promotion, and this portion is affected via different strategies such as crop rotation, green manuring, and sustainable use of PGPM and intercropping. However, through the addition of PGPM, Asghar and Kataoka [33] found that plant growth-promoting fungi changed the fungal community composition, enhanced the phosphatase and glycosidase enzyme activities in the soil, and stimulated the lettuce and brassica plant growth. Further, Kataoka et al. [34] reported that incorporating leguminous green manure (Hairy Vetch, *Vicia villosa* L.) enhances the soil fungal biomass and diversity and provides a healthier environment by producing soil phosphatase enzymes. Furthermore, rhizosphere engineering and its processes are explained with different strategies. In short, these interventions improve the rhizospheric processes and sustainably increase the agroecosystem and its productivity. On the other side of the agroecosystems and maintaining the rhizosphere process, biochar has emerged as a crucial innovation in scientific research, offering multiple benefits for sustainable agriculture, rhizosphere management, and environmental health [35]. Biochar is a type of charcoal produced through the combustion of organic feedstock with limited oxygen. It is considered a valuable soil conditioner and an effective means of carbon sequestration, which helps mitigate global warming and climate change [36]. Biochar decomposes slowly, making it highly durable when applied to soil, where it can remain for the long term [37]. However, generally, biochar is derived from waste materials such as animal manure, forest residues, and agricultural byproducts [38]. Through scientific processes, these waste materials are transformed into valuable products that can directly or indirectly improve soil health and promote plant growth. Biochar enhances crop yield, improves soil quality, and maintains soil health by preserving biochemical properties [39]. In terms of soil biological properties, Uzoma et al. [40] found that biochar positively influences soil microbes and their activity, leading to increased crop yields. However, the effect of biochar depends on the specific soil type and specific crops. Another study by Lu et al. [41] highlighted that biochar’s porous structure creates a favorable niche for soil microbiota, boosting microbial populations. Conversely, biochar derived from rice straw has been shown to reduce certain microbial communities, such as *Ascomycota* fungi and *Actinobacteria*, though the abundance and diversity of soil microbes may vary with different types of biochar [42]. The application of biochar not only alters soil microbial communities but also affects nutrient cycling, which can have a direct impact on rhizospheric processes and agroecosystem productivity.

## 3. Role of Soil Microbes in the Rhizosphere Processes

The rhizosphere is home to various soil microbes, including fungi, bacteria, protozoa, and nematodes, which play essential roles in nutrient cycling, plant growth, and maintaining agroecosystems. These microbes facilitate the breakdown of organic matter and nitrogen fixation, making nutrients available to plants [4]. In agriculture, two groups of soil microorganisms are particularly important—rhizosphere fungi and bacteria—due to their critical functions. Fungi and bacteria thrive in soil, adapting to a wide range of environments, including harsh and high-salt conditions [43,44]. Fungi in the rhizosphere include pathogenic, saprotrophic, and mutualistic species, with mycorrhizal and plant-growth-promoting fungi being especially important for increasing nutrient availability, promoting plant growth and stress tolerance, and improving soil structure and disease resistance [45]. In contrast, the main rhizosphere bacterial phyla are *Firmicutes*, *Actinobacteria*, *Acidobacteria*, and *Proteobacteria* [46]. Plant growth-promoting rhizobacteria (PGPR) are crucial for enhancing plant growth, improving disease resistance, responding to carbon inputs and nutrient uptake, and assisting in phytoremediation [47,48,49]. In recent decades, a substantial body of literature has emerged demonstrating the activities of microbial species in plant growth promotion, biocontrol of phytopathogens, and nutrient cycling, highlighting their potential for development as alternative or supplementary agrochemicals in sustainable agriculture.

### 3.1. The Role of Microbes in Agroecosystems

Rhizosphere microorganisms, such as fungi and bacteria, not only act as pathogens in agricultural plants but also facilitate plant growth, increase nutrient availability, and control plant diseases through their role as biocontrol agents [50]. Recent research has highlighted the utilization of rhizosphere microorganisms to enhance plant development and mitigate diseases, thereby reducing reliance on pesticides and herbicides [51]. These microorganisms establish symbiotic associations with plant roots, exemplified by mycorrhizal fungi, which augment water and nutrient absorption, especially phosphorus. Furthermore, specific bacterial and fungal species might inhibit detrimental infections, thereby reducing dependence on chemical pesticides. Furthermore, these microbes help in the degradation of pollutants, thereby improving soil quality and resilience [49,52]. In comparison, the excessive application of chemical fertilizers and pesticides has detrimental environmental consequences, and growing reliance on these substances is concerning. Commercially available as biofertilizers and biocontrol agents, a wide range of microorganisms are utilized in agricultural production and the management of agroecosystems through rhizosphere processes that chemical fertilizers often disrupt. The diversity and activity of soil microbial communities are essential for preserving soil structure, enhancing plant growth, and sustaining ecosystem stability, rendering these microorganisms crucial for sustainable agricultural operations [53]. Therefore, enhancing the resilience of agroecosystems through rhizosphere microbes is crucial for promoting sustainable agriculture.

### 3.2. Plant Growth Promotion via Microbes

The application of chemical fertilizers to meet the food demand for a growing human population has led to many unpredicted environmental concerns related to soil–plant health and agroecosystems [54]. To protect soil–plant health and the environment, there is a need to develop and adopt new biological techniques that are not harmful to agroecosystems, soil, and plant health. Many researchers and scientists have already reported that microbes related to biofertilizers are functional and could be an alternative to chemical fertilizers, provide maximum environmental benefits, and improve crop yield and soil microbial biomass. In this context, microbe-related biofertilizers are capable and could be used in an eco-friendly manner to enhance crop production and maintain agroecosystems [55]. Microbe-related biofertilizers are defined as biological inoculants of beneficial fungal and bacterial strain mobilizers that can increase crop productivity and sustain soil biodiversity [56]. Mazid and Khan [56] reported that these inoculants, applicable directly or indirectly to soil and plant roots, are cost-effective and supply important nutrients for plants, enhancing soil nutritional composition, structure, and fertility.

Moreover, biofertilizer-related microbes are commonly known as plant growth-promoting microbes (PGPM). There are various fungal and bacterial strains that have already been reported and used commercially to improve plant growth, soil conditions, and the nutritional demand of plants. Such bacterial and fungal strains are *Azotobacter*, *Pseudomonas*, *Azospirillum*, *Rhizobium*, *Streptomyces*, *Bacillus* spp., *Trichoderma*, *Penicillium*, *Phoma*, *Fusarium*, and *Aspergillus* spp., and these strains have been gaining global attention due to their potential to control soil diseases, promote plant growth, and maintain agroecosystems [57,58,59]. The significant role and potential activities of PGPM are shown in Table 1.

### 3.3. Singling Between Plants and Microbes in the Rhizosphere

The rhizosphere contains four main components: soil, microorganisms, roots, and their interactions. These components collectively influence a range of physical, chemical, and biological processes that affect nutrient use, plant development, and overall plant health. The primary goal of rhizosphere signaling is the interaction between plants and the symbiotic microbes in their surrounding area, which promotes the growth of various rhizosphere microbial communities that have a beneficial impact on plant productivity [70]. A robust reciprocal influence marks the interaction between the plant and the rhizosphere microbiome (Figure 1), where they interact through the exchange of signaling molecules produced and recognized by both the plant and its related microbiosystem [71]. Disease-free and asymptomatic plants have complex relationships with their rhizosphere microbes, which improve plant performance and maintain the agroforestry system [72,73]. Plants change the pH level, the structure of the soil, the amount of oxygen that is available, and the energy source that comes from carbon-rich exudates. These changes have an effect on the microbiome around the plants’ roots. Many of the chemically varying primary and secondary metabolites found in plant root exudates have bioactive effects on microorganisms, influencing their composition and function [74]. The exudation of carbon from plant roots, which accounts for almost one-third of the carbon produced by photosynthesis, significantly influences the results of chemical interactions at both the individual and community levels [75]. Root exudates serve as a principal means of communication between plants and their living environment, enabling several reactions, including nutrient uptake, competition for resources, signaling across species, the attraction of microorganisms, and several other interactions [76]. Furthermore, sugars, amino acids, organic acids, phenolic compounds, and secondary metabolites like coumarins, glucosinolates, benzoxazinoids, camalexin, and triterpenes are the main organic parts that come from exudate [77,78]. Plant species cultivate a unique microbial community in their rhizosphere by creating a varied carbon-rich environment. With more evidence, primary metabolites, including glucose, organic acids, and amino acids, supply carbon and energy for microbial proliferation, promoting beneficial microorganisms and inhibiting pathogens. Secondary metabolites, like flavonoids, phenolics, and terpenoids, often have bioactive properties, like the ability to kill bacteria or transfer information, that affect only the microbes in the soil and plants, and these changes in microbial community structure can significantly impact soil and agroecosystem [79,80,81]. This community provides many adaptive benefits to the plant host by influencing the composition of microorganisms and adjusting their advantageous characteristics [78]. Additionally, chemicals made from root exudates play a key role in creating a stress-resistant microbiome that helps plants deal with abiotic stresses like not getting enough food, being sick, and drought. The identification of stress-derived metabolomics and microbiota is a viable approach to addressing both abiotic and biotic limitations [82,83,84]. However, we have not thoroughly investigated the positive impacts of root-enriched microbial species supported by specialized metabolites produced from root exudate [85]. This paragraph of the manuscript on a review of current research on the plant rhizosphere microbiota and the compounds produced by root exudates for nutrient acquisition as depicted in the conceptual (Figure 2).

## 4. The Role of Green Manures in Agroecosystems

Green manures represent a promising strategy for sustaining agroecosystems and improving the soil health and rhizosphere processes. Green manure can reduce chemical fertilizers in agroecosystems, ultimately improving soil health and quality [86]. Green manure provides various benefits to the agroecosystem and soil, such as soil covering, reduced soil temperature, organic matter improvement, increased water infiltration, and reduced weed infestation [87]. Moreover, green manure has environmental and economic benefits such as food production, maintaining biodiversity, soil carbon sequestration, and soil retention [88]. However, to enhance the resilience of agroecosystems and maintain soil health, planting green manure is an essential and valuable practice and works to reduce soil-borne diseases through biofumigation. Green manure provides better soil health and biodiversity and promotes the beneficial fungal genera that break the soil-borne pathogen’s life cycle that is connected with the crop genotype [89]. Many types of green manures, including leguminous and non-leguminous, have multiple functions and benefits for soil and plant health, such as incorporating leguminous green manure to enhance wheat yield and production due to the mineralization of inorganic nitrogen and stimulating biological fixation [90]. Furthermore, Kataoka et al. [34] also reported that incorporating leguminous green manure (Hairy Vetch, *Vicia villosa* L.) enhances the soil fungal biomass and diversity and provides a healthier environment by producing soil phosphatase enzymes. Therefore, it could be assumed that planting and incorporating different green manures could maintain the agroecosystems and rhizosphere processes by enhancing biological processes and increasing organic matter. From another point of view, green manure may alter the soil N and C availability. For example, Hairy Vetch (a leguminous plant) fixes N_2_, whereas barley (a non-leguminous plant) has high biomass productivity and thus adds more C, which influences ecosystem functioning and microbial groups in soil [91]. In comparison, Chavarría et al. [91] reported that green manure incorporation can improve agroecosystem services by improving nutrient retention capacity, accelerating soil organic matter, and reducing greenhouse gases in agricultural soils. However, applying alternative management, such as green manuring practices, stimulates the improvement of soil-diverse microbial communities, accelerating the specific enzyme activities related to P and C [92]. Thus, the relationship between microbial community composition and their activity stimulates plant growth and soil quality.

### 4.1. Green Manure and Their Beneficial Effects on the Rhizosphere Processes

The application of green manure is an additional practice for enhancing the resilience of agroecosystems and rhizosphere processes and increasing soil’s productive capacity through nutrient availability. This application is geared towards maintaining agroecosystems and long-term environmental sustainability by enhancing processes such as nutrient cycling and preserving and maintaining soil biodiversity and the physical and chemical stability of the soil. Incorporating green manure can positively affect soil microbial communities by directly adding nutrients after incorporation and indirectly by changing plant and soil properties [93,94]. Directly, the decomposition of green manure offers an immediate supply of carbon, nitrogen, and other vital elements; hence, it enhances microbial activity and fosters the proliferation of beneficial bacteria [95]. The mineralization of organic matter from green manure produces labile chemicals that act as substrates for soil microorganisms, thereby boosting microbial biomass and diversity [96]. The incorporation of green manure indirectly modifies the soil’s physical and chemical characteristics, including pH, soil structure, and water retention, fostering a conducive environment for microbial colonization and activity [97]. At the same time, Sun et al. [45] reported that the plant could affect soil microbial diversity and community by releasing root exudates. The decomposing residues of brassica green manure release glucosinolates, which assist in controlling parasitic nematodes [98]. Furthermore, the exclusive use of grass species may improve the soil nutrient profile compared to monoculture. The advantages of green manure designate it as a sustainable practice capable of providing various agro-ecological services; however, effective management of green manure is essential to substantially enhance soil carbon stocks and mitigate climate change, particularly in semi-arid Mediterranean regions, by decreasing greenhouse gas emissions [99,100]. Furthermore, many studies have proven that planting and incorporating green manure have multiple beneficial effects on soil health and the rhizosphere process [86,101]. The different types of green manure and their beneficial effects on the rhizosphere are shown in Table 2.

### 4.2. Green Manure and Their Effects on Soil Health

Soil health is a vital component of sustainable agriculture, signifying the soil’s ability to operate as a living ecosystem that promotes plant productivity, preserving environmental quality, and supporting biodiversity [108]. Healthy soils exhibit an appropriate combination of physical, chemical, and biological attributes, encompassing adequate organic matter, nutrient accessibility, optimal structure, and vibrant microbial communities. Beneficial microorganisms, including nitrogen-fixing bacteria, mycorrhizal fungi, and decomposers, are vital to nutrient cycling, organic waste breakdown, and disease suppression [109]. Soil health is a primary requirement for agriculture and plant growth, but most agricultural soils are poor in organic nutrients and organic matter due to the high applications of inorganic or chemical-based fertilizers to achieve high crop yields. Different agricultural techniques should be adopted; otherwise, it can become very hazardous and lead to a lack of organic matter. Improving soil health through chemical, physical, and biological properties is attributed to the integrated application of green manure and organic application due to better nutrient uptake and preservation of soil health [110]. A variety of green manures, including grains, root crops, legumes, and oil crops, can serve as vegetative covers, each offering different benefits for enhancing soil health. Therefore, growing and incorporating green manure in agricultural lands helps maintain and preserve soil health and soil nutrient availability and ultimately increases soil organic matter content. The incorporation of green manure provides soil nutrients and organic matter to the soil, and organic matter plays an important role in soil biodiversity, soil microbiota, and the arrangement of soil aggregates to improve the soil structure [111,112]. Therefore, incorporating green manure leads to an increased organic matter content, which has an important role in soil health and quality. Furthermore, green manure suppresses soil-borne diseases through biofumigation. Soil-borne diseases also affect soil health, reduce crop productivity, and negatively affect soil biodiversity. Many soil-borne pathogens have negative effects on plant growth and crop production, such as *Phytophthora* spp., *Fusarium* spp., *Verticillium* spp., and *Rhizoctonia* spp. Soil-borne diseases reduce agricultural yield by as much as 50–70%, including those of wheat, vegetables, and cotton [113,114]. However, the practices of green manure in agricultural lands produce significant benefits such as increased soil nutrient availability, enhanced soil organic carbon, reduced soil compaction, improved particle aggregation and soil structure, as well as strengthened microbe activity, diversity, abundance, and suppression of soil-borne diseases.

To protect crops from soil-borne diseases, many fumigants and fungicides need to be used regularly. Later, it was noted that the use of fumigants and fungicides or any other chemical-based product causes ecological problems such as human health hazards and reduced beneficial microbes in the soil biodiversity, directly affecting soil health [115]. Consequently, organically, alternative strategies should be adopted to reduce soil-borne diseases, preserve soil health, and improve crop production. Alternatively, incorporation of green manure and cover crops are useful strategies for protecting soil health from soil-borne diseases and improving soil and plant health through biofumigation, providing nutrients and increasing beneficial microbial activity. Hao et al. [116] reported that broccoli (*B. oleracea*) grown as green manure reduces the *S. minor sclerotia* in the rhizosphere. Whalen [117] also reported that Fusarium wilt is another soil-borne disease that reduces watermelon production in the southeastern United States. The soil-borne disease Fusarium wilts is due to *Fusarium oxysporum*, which attacks the crop of watermelon and reduces the annual number of fruits, directly affecting the annual yield of watermelon production. For the management of Fusarium wilt disease, Hairy Vetch (*Vicia villosa* L.) green manure was incorporated. After incorporating Hairy Vetch, the occurrence of Fusarium wilt disease in the rhizosphere of the watermelon field was reduced, leading to an increased annual watermelon yield by 45% [118]. Papp et al. [119] demonstrated that the cultivation of green manures influences soil microorganism environments, which are essential for sustaining soil functions and ecological system sustainability because they participate in nutrient cycling and organic matter turnover, providing multiple services. Researchers have observed that the incorporation of green manure modifies the composition and dynamics of soil fungus and bacterial communities and encourages beneficial microorganisms. Ntalli and Caboni [120] observed that certain cruciferous green manures, including canola (*Brassica napus* L.) and rape (*Brassica rapa* L.), upon termination, release isothiocyanates (ITCs) via the hydrolysis of glucosinolates (GSLs) during the decomposition of their tissues, serving as effective biofumigants against various soil-borne pathogens and pests. The utilization of cruciferous species as green manure may facilitate the natural management of possible soil-borne diseases. Furthermore, Waisen et al. [121] demonstrated that brown mustard (*Brassica juncea* L.) and radish (*Raphanus raphanistrum* subsp. sativus) serve as excellent biofumigant crops that protect against plant–parasitic nematodes while keeping the soil healthy and maintaining the strength of the nematode population structure strong. Aydınlı and Mennan [122] also found that planting arugula (*Eruca sativa*) and radish (*R. sativus*) as winter crops before planting plants that are easily damaged by the root-knot nematode Meloidogyne arenaria may lower the number of eggs laid, the gall index, and the damage that occurs, while also increasing crop yields. Leguminous green manure can substantially enrich the soil with nitrogen throughout its growth and can acidify the rhizosphere by enhancing the absorption of insoluble phosphorus into the soil [123,124]. Leguminous green manure can fix atmospheric nitrogen via symbiosis with rhizobia in root nodules [125]. Furthermore, the fixed nitrogen can be allocated to intercropped non-leguminous plants within mixed cropping systems, or it may succeed crops in rotational practices. Biological nitrogen-fixing (BNF) systems may decrease the reliance on commercial nitrogen fertilizers [126,127]. Further, Scavo et al. [128] reported that the existence of *Trifolium subterraneum* for three straight years resulted in a significant rise in nitric nitrogen, ammoniacal nitrogen, and nitrogen cycle bacteria. Campiglia et al. [129] observed similar wheat yields when they used underground clover as living mulch in intercropping systems. In addition, Guardia et al. [130] showed that the mitigating effect of the legume green manure (vetch) mostly comes from less synthetic nitrogen being added to the next cash crop, less indirect N_2_O emissions from NO_3_^−^-leaching, and more carbon being stored because photosynthesis is higher.

More specifically, green manures are essential for improving soil health via a mix of physical, chemical, and biological processes that promote sustainable agroecosystems. The integration of green manure biomass augments soil organic matter, thereby enhancing soil structure, increasing water retention, and facilitating aeration, which fosters optimal circumstances for root growth [95,96]. This organic supplement serves as a nutrient reservoir, progressively releasing vital macronutrients such as nitrogen, phosphate, and potassium throughout decomposition and thus diminishing dependence on synthetic fertilizers. Green manures increase the diversity and activity of microbes in the soil. They do this by creating a dynamic food web that supports good microbes and stops soil-borne diseases through competitive exclusion and antibiotic synthesis. Furthermore, green manures facilitate carbon sequestration by integrating plant-derived carbon into stable soil organic components, thus reducing greenhouse gas emissions and enhancing soil resistance to climate change [131,132]. Adding leguminous green manures also helps fix nitrogen, and the breakdown of bioactive chemicals like glucosinolates in cruciferous green manures acts as natural biofumigants, reducing the number of pests and diseases that can affect the crop. Integrating green manures into cropping systems enhances soil health and bolsters agroecosystem sustainability, establishing it as a fundamental practice for climate-resilient agriculture [18]. Therefore, green manure and cover crops should be adopted to protect soil health, reduce soil-borne diseases as well as refine water use efficiency and many more beneficial impacts, as depicted in the conceptual (Figure 3).

## 5. Role of Cropping Systems in Agroecosystems

Understanding the value of agroecosystems requires knowledge of soil, its different properties, and the processes that maintain its health. Unfortunately, the soil is currently being degraded rapidly due to extensive application of chemical-based fertilizers, lack of agricultural techniques, and irregular cropping are rapidly degrading the soil, but what is most concerning is that soil is a non-renewable resource at a human temporal scale [89]. Therefore, scientists and researchers have been trying to adopt alternative strategies to protect soil health, ultimately enhancing agroecosystems and rhizosphere processes. Cropping systems, such as intercropping and crop rotation practices, could serve this purpose in agricultural practices, thereby positively influencing soil health and agroecosystems [133]. Cropping systems are adopted to maximize the crop yields of agroecosystems as well as to maintain agroecosystems through soil health and quality preservation. In contrast, the minimization of chemical-based products directly or indirectly affects agroecosystems and rhizosphere processes [134]. With the primary goal of soil health maintenance to ensure the long-term stability of agroecosystems and the high production of crops, we need to adopt agronomic practices such as intercropping or crop rotations. Intercropping practices can improve agroecosystems by reducing the use of chemical-based fertilizers and soil pollution, preserving soil microbial diversity.

### 5.1. Intercropping and Their Effects on Rhizosphere Processes and Soil Health

Intercropping, which involves the cultivation of two or more crops in a field concurrently, is a useful strategy in agronomic practice. Different crops, such as cover and cash crops, grow at the same time. The practice has gained global attention due to its beneficial effects on the enhancement of agroecosystems and soil health. In brief, intercropping practices can maintain agroecosystems and soil health by the reduced use of chemical-based fertilizers [135], enhancing soil-plant functions [136], suppressing the occurrence of soil-borne diseases [133], increasing soil nutrient and organic matter contents [137], and promoting the beneficial soil microorganisms, which support the rhizosphere processes [138]. Cong et al. [139] reported that intercropping systems, including that of faba beans, corn, and wheat, had about 11%, 4%, and 23% higher below-ground plant biomass as well as increased organic carbon (C) and nitrogen (N) contents than those in plant species rotations found in Gansu, China. Furthermore, de Medeiros et al. [140] reported that intercropping with beans and pigeon peas considerably minimizes black root rot (*Scytalidium lignicola*) in cassava by up to 50% when matched with cassava in monoculture systems, as well as enhanced the rhizosphere process, that is, increasing the soil enzymes activities, microbial biomass, and soil nutrients related to organic C and N. Increasing the use of intercropping with different plant species have been typically interconnected with the enhancement of rhizosphere processes, providing environmental benefits [141]. For example, leguminous species are advantageous for soil by providing nitrogen-fixing microorganisms and enhancing soil enzyme activities. On the other hand, non-leguminous species usually provide soil cover and nitrogen availability; such intercropping self-regulates soil nitrogen levels to improve soil nutrient use and reduce the C footprint [142]. Furthermore, Graf et al. [143] also reported that compared to sole cropping, intercropping practices on *Dactylis glomerata* and *Medicago sativa* have led to increased shoot biomass and nitrous oxide (N_2_O) production rate, which suggests that understanding the intercropping strategies is helpful for the maintenance of agroecosystems and soil health. Intercropping is regaining agronomical and ecological importance because of its ability to self-regulate soil nutrients and preserve soil health. An overview of intercropping in agroecosystems and its beneficial interactions based on Brooker et al. [144] and their outcomes are shown in Figure 4.

Furthermore, legume and cereal intercropping systems have been extensively employed to address limitations on natural resources, promote sustainable agricultural development, and guarantee food security due to their economic and ecological advantages [145]. Faba/wheat intercropping decreased nitrogen fertilizer application by 5–16% while simultaneously enhancing wheat production by 15–25% [146]. However, Luo et al. [147] reported that a 40 percent reduction in synthetic nitrogen input, coupled with soybean intercropping, might sustain sugarcane productivity and promote sustainable soil management. Many research studies have demonstrated that below-ground facilitative interactions among species can improve nutrient availability and utilization. Ref. [148] revealed that the root interactions between maize and faba beans, along with the root exudates from maize, significantly improved the symbiotic N_2_ fixation and nodulation in faba beans grown alongside maize. Intercropping can make crop roots release more organic acids and control the mineralization of organic phosphorus, the dissolution of inorganic phosphorus, or both. This makes more phosphorus available in the soil. Recent research shows that growing cereals and legumes together improves metabolic function and changes the structure of the soil’s microbial community through interactions between species [149]. According to Zhang et al. [150], elevated carbon uptake by maize (*Zea mays* L.) promoted nutrient cycling by altering the abundance of functional groups in soil microbes and improving the stability and complexity of microbial networks in soybean/maize intercropping systems. Teshita et al. [151] proposed that, particularly in low-nitrogen soils, intercropping maize with alfalfa (*Medicago sativa* L.) could enhance the structural complexity of the soil food web. Moreover, intercropping improves agricultural sustainability by utilizing crop diversity to exploit synergistic interactions across species [152]. This method improves crop yield and nitrogen utilization efficiency through similar and beneficial interactions. The rhizosphere of leguminous plants cultivates distinct microbial communities, including nitrogen-fixing bacterium (nifh) populations, via root exudates in legume-based intercropping systems, enhancing ecological benefits such as nitrogen fixation and phosphorus mobilization [153,154]. Within these approaches, maize/soybean intercropping is notable for its effective transfer of fixed nitrogen from soybean to maize, hence improving nitrogen efficiency and yields [155]. Multiple factors, such as cropping configurations and fertilization methods, affect how well maize and soybean intercropping systems work. Improving microbes is crucial for nutrient absorption and yield maximization [156,157]. Understanding how rhizosphere microbial communities react, especially in terms of their ability to fix nitrogen, would help improve intercropping patterns and fertilizer management, which would lead to more sustainable farming and food security.

### 5.2. Crop Rotation and Their Effects on Rhizosphere Processes and Soil

Crop rotation is a conventional and effective method for maintaining agroecosystems, managing biodiversity by improving soil health, suppressing disease and pest occurrences, and hence enhancing crop yields [158]. The efficiency and value of crop rotation are dependent upon various elements, including the types of crops utilized in the rotation, the sequence and frequency of specific crops, the duration of rotation, the agronomic history of the agricultural land, and the properties of the soil [159,160]. Recent studies reported that farmers had used crop rotation through traditional methods to manage crop pests and diseases and manage soil productivity [89,161]. Usually, these crop rotations involve growing three or four different types of crops in a sequence. However, with increasing food demand and agricultural production in recent years, many farmers grow just one or two crop species, with pesticides and mineral fertilizers contributing to compensate for the lack of crop rotations [162]. The crop rotations are not always helpful and also decline the soil quality, and crop production has been reported with the number of crops grown linked with short rotation or continuous cropping, including soybean, sugarcane, maize, wheat, and oilseed rape [163]. Interestingly, in crop rotation, oilseed rape is very important for the rotation and even highly profitable when grown as a break crop for cereals. Crop rotations with oilseed rape and maize affect the dynamics and structure of soil plant-associated microbial communities and are known to be beneficial for soil health via suppression of soil-borne diseases and plant growth promotion [164]. Microbial species living nearby or associated with plants are directly influenced by the root’s architecture and the chemical characteristics of root exudates when we rotate with maize or oilseed rape crops [165]. The direct influence of plant roots around soil is known as the rhizosphere. The rhizosphere is the habitat of plant-beneficial microorganisms and plant pathogens [109]. With the research evidence of Benitez, Osborne, and Lehman [164], crop rotations promote beneficial plant growth-promoting microbes, affect the soil microbial community in the rhizosphere, and influence maize seedling growth characteristics.

Furthermore, rotating crops with grain legumes can significantly improve the yield and protein content of subsequent wheat crops, owing to the increased nitrogen availability in the soil from legume biological fixation [90]. Leguminous crops (such as peas and chickpeas) and various chickpea genotypes (cultivars) in rotation can alter soil functional microbial populations and affect the productivity of pulse crops and subsequent wheat crops. Different crops can produce root exudates and distinct residues that enhance soil microbial diversity and activity, as well as microbial biomass, nitrogen, and carbon cycling [160,166,167]. Modifications in crop rotation duration and frequency of identical crops throughout time might influence the prevalence of root rot diseases and improve crop production and soil health. Short series rotations exhibit heightened sensitivity to host-specific diseases, resulting in poorer yields compared to lengthier series rotations [162,168]. In addition, Lupwayi et al. [169] found that the wheat phase of a five-year rotation exhibited greater microbial biomass in both rhizosphere and bulk soil compared to the three-year rotation of the wheat phase, likely due to increased carbon inputs from crop residues. In Canada, particularly in the western part, two phases of pea in a four-year rotation increased soil nitrogen levels, whereas three legume phases markedly altered the function and composition of the rhizosphere bacterial community in comparison to continuous wheat cultivation and growth [170]. Increasing the rate of one crop in rotation can be detrimental to soil health. For example, Bainard et al. [171] observed that a higher pulse phase in rotation caused host-specific fungal pathogens to build up in the soil, which could make rotation less beneficial for crop yield and soil health. Meanwhile, an additional crucial factor in crop rotation design is the ability of soil-borne diseases to utilize alternate crops as hosts or to remain dormant in the soil for extended periods, as well as the response of these crops to disease. Utilizing non-host plants in crop rotations to manage diseases transmitted by the soil is critical for mitigating yield losses, particularly given that many pathogens can persist in the soil for extended periods as spores or other latent forms in the absence of their preferred host plants [162,172]. For example, Nayyar et al. [173] reported that Fusarium root rot in peas cultivated in rotation on the Canadian prairie was associated with a restricted soil microbial community and reduced populations of arbuscular mycorrhizal fungi and beneficial bacteria. In certain instances, continuous cropping with enhanced crop diversification elevated the population of antagonistic soil microorganisms, hence lowering the populations of soil pathogens and reducing the “take-all” effect in wheat. Typically, incorporating three or more crops within a cropping system can improve soil health and optimize the productivity of crops [89,162]. However, the benefits of crop rotation are practical; understanding the specific effects of crop systems and how they affect subsequent crops will encourage and adopt a worthy crop rotation, which will promote soil health and increase crop productivity. These findings make it easier to create customized interventions that use microbial populations to maintain soil health, decrease the emissions of greenhouse gases (GHGs) and crop production, and the efficient use of resources in intercropping systems. According to references, we propose intercropping in agroecosystems, as depicted in the conceptual (Figure 5).

## 6. Indicators for Assessing Soil Health in Agroecosystems

Soil health typically refers to the capacity of soil to operate as a crucial living system that supports biological productivity, preserves both soil and plant health, enhances environmental quality, and maintains an agroecosystem [174]. Soil is a highly intricate and complex, multifunctional system including gaseous, liquid, and solid elements that interact through various chemical, physical, and biological processes. Healthy soil sustains agroecosystems and facilitates the provision of essential services to ecosystems [175]. In order to evaluate and sustain agroecosystems, it is essential to consider the biological, physical, and chemical functions and traits of the soil, particularly the biological ones that serve as sensitive indicators of soil health and quality [176]. Microbiological and biochemical markers show that the variety and activity of soil microbes are important for the long-term health of agroecosystems because they keep soil health functions going, like cycling carbon and nutrients [177]. Microbial indicators exhibit more sensitivity than physical traits to environmental alterations such as soil usage and management; hence, they enable early detection and forecasting of changes and disturbances in environmental sustainability [178]. Related studies recognize soil microbial biomass, which includes fungi, bacteria, and algae, as a primary biological indicator of soil health and as a crucial source of nutrient cycling and delivery relative to plant demand [179]. Additionally, through photosynthesis, plants fix and transfer carbon as carbohydrates into the food web, making it one of the most critical biological processes on Earth [177]. Both agricultural and forestry soils widely use soil respiration and microbial biomass as bio-indicators of soil health.

We expect biochemical indicators to effectively integrate the combined impact on soil’s chemical, physical, and biological processes and characteristics, making them suitable for diverse management and environmental conditions. This paragraph aims to elucidate and augment our focus on soil quality and related research, particularly with biochemical soil quality indicators such as enzymatic and microbial activity [180]. Due to their role in nutrient cycling through microbial processes, soil enzyme activities frequently serve as sensitive indicators of soil biochemical quality and as measures of ecological quality. They can effectively respond to both anthropogenic and natural alterations in soil and are easily quantifiable [181]. However, soil enzymes are important in the functioning of soil ecosystems, including nutrient and carbon cycling and maintenance [182]. For example, soil enzymes have an important role in C (*β*-glucosidase and galactosidase), N (urease), and P (acid phosphatases and alkaline) cycling, and these enzymes play crucial roles in the breakdown of OM and release nutrients, which are important for soil health and plant growth promotion. Dick and Tabatabai [183] reported that soil enzymes serve as reliable indicators of soil quality because of their close association with soil biological processes, their ease of measurement, and their swift response to alterations in soil management. In summary, soil enzymes are valuable tools for assessing both long and short-term changes in soil quality and management practices.

Furthermore, indicators of soil health for sustaining agroecosystems need to be associated with soil processes and responsive to alterations in management and environmental conditions. Soil biological traits, such as microbial biomass and its activities, serve as sensitive and rapid indicators that capture responses and data from diverse environmental conditions [184,185]. We use soil biological metrics, such as microbial density, activity, and biomass, as indicators of soil health [186]. Furthermore, a specific drawback of employing soil microbial characteristics as soil health indicators is the technological limitations encountered in the research of soil microbial populations. Molecular techniques that are more advanced, especially next-generation sequencing (NGS) methods (such as shotgun sequencing for structural and amplicon sequencing and functional microbial diversity investigations), have made it possible to study the part of soil microbial communities that cannot be grown in a lab. This is because most soil microorganisms do not grow on media or in a lab setting. However, while analyzing the data, we must acknowledge the limitations of these novel methodologies. In conclusion, we can use soil to focus on agroecosystem maintenance and enhance rhizosphere processes; soil health indicators are used to inform management techniques, optimize fertilizer application, and promote sustainable crop productivity.

## 7. Integrated Agricultural Practices for Improving Rhizospheric Processes

Integrated agricultural practices are essential for improving rhizospheric processes, which are important for sustainable crop yield and soil health. Integrated agricultural practices established soil properties (biological, chemical, and physical properties) that influence nutrient cycling and soil microbial community composition and structure and enhance the rhizospheric process [187]. However, to reduce adverse effects, integrative biofertilizers, crop rotation, green manuring, intercropping, and mulching practices should be promoted and tested worldwide to improve the rhizosphere processes and agroecosystems resilience. The disturbance of rhizosphere processes and depletion of organic matter are restraining agricultural productivity around the globe via improper agricultural practices. In short, integrated agricultural practices (IAP) are based on the integration of different crops, livestock wastes, and fertilizers into production systems that, through valuable management practices, maintain a high level of soil fertility, productivity, and quality, reduce external inputs of agrochemical and fertilizers, and enhance the functions of soil inside biological cycles and its processes. Mixed-crop livestock and other available resources, such as seed primed—the basis of integrated agricultural practices—allow the most efficient and effective use of natural resources and biological cycles to improve below-ground activities [188]. Sarkar et al. [189] reported that microbial-assisted nutrient management is a sustainable, eco-friendly, and cost-effective option under integrated agricultural systems. The usage of less amount of agrochemicals provides support organically or naturally to protect the environment from nutrient denitrification or runoff, land degradation, and soil pollution. Duarah et al. [190] stated that reducing the amount of synthetic NPK fertilizer application and seed priming (with bacteria) can significantly improve nitrogen-use efficiency. However, Entesari et al. [191] witnessed that bio-priming seeds of soybean with *Trichoderma* sp., prior to planting, enhanced the nutrient status of the crop and improved the crop yield. Phosphorus (P), an essential nutrient, is often fixed in the soil and forms complexes with metals, limiting its availability to plants. However, certain fungal and bacterial communities have a trait called P solubilization, which allows them to convert unavailable or fixed phosphorus into forms that plants can absorb. Kim et al. [192] reported that incorporating green manure can help re-mobilize phosphorus and make non-exchangeable potassium available, circulating nutrients more efficiently. This integrated approach significantly enhances phosphorus availability, a key element for plant growth and the activity of soil microorganisms. Integrated Agricultural Practices (IAP) provide numerous advantages by carefully selecting crops and utilizing available resources to develop creative, sustainable systems. These practices effectively achieve several critical objectives: minimizing disease and insect problems, reducing inputs and energy requirements, and lowering the reliance on agrochemicals and pesticides. Additionally, IAP mitigates risks associated with climate variability and economic fluctuations, making agricultural systems more resilient. Improved rhizosphere processes foster healthy soils, which are vital for sustaining ecosystem services that benefit plants, animals, and humans alike. By optimizing rhizospheric functions, agroecosystems can better support nutrient cycling, enhance soil biodiversity, and maintain productivity, ensuring the sustainable production of food and fiber. Ultimately, this integrated approach is essential for strengthening agroecosystem resilience, safeguarding long-term agricultural sustainability, and addressing the challenges of future global food security.

## 8. Concluding Remarks and Future Directions

Based on the evidence, recognizing the importance of rhizosphere processes and their interactions with microorganisms is essential for promoting sustainable agroecosystems. The rhizosphere fosters various microbial populations that augment nutrient accessibility, regulate pathogens, and promote soil structure. Some important things that must be performed to rebuild microbial communities, improve soil health, and make nutrient cycling easier are making green manure, intercropping, and using microbial inoculants. These solutions diminish dependence on synthetic fertilizers, augment carbon sequestration, and improve plant resilience to abiotic stress, fostering steady agricultural yields and economic advantages. Optimizing rhizosphere functions enables agroecosystems to enhance food and fiber production, preserve soil fertility, and ensure long-term sustainability, fostering resistance to environmental challenges and safeguarding global food security. However, the findings from this review provide valuable insights to agronomists, soil scientists, and crop managers on optimizing rhizosphere processes to improve agroecosystem resilience. In closing, the rhizosphere will be central to future ecological innovations in agriculture, driving both resilience and sustainability. However, further experimental research is required to fully unlock the prospect of the rhizosphere. Future studies should focus on integrating advanced technologies, such as microbial genomics and precision agriculture, to optimize rhizosphere management for more resilient and sustainable agroecosystems. The development of smart agricultural strategies, coupled with policy support, will be vital for addressing the growing challenges posed by climate change and soil degradation.

## Figures and Tables

**Figure 1 ijms-26-00109-f001:**
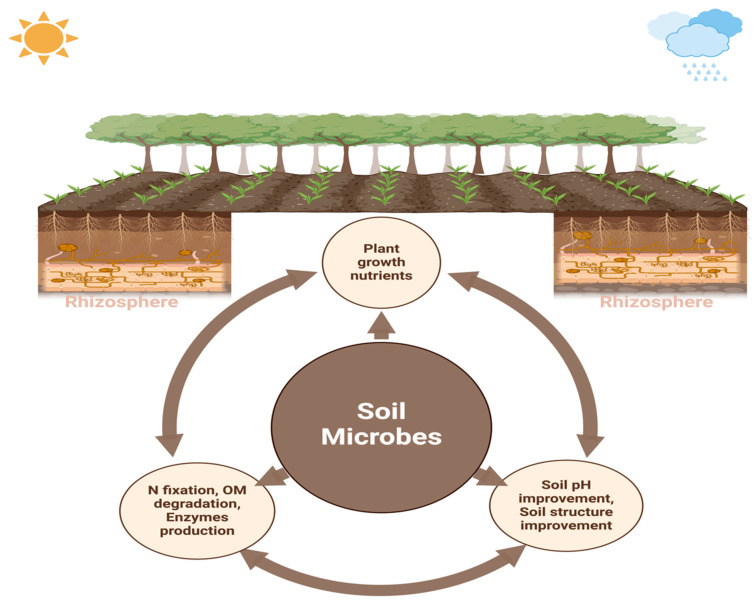
The rhizosphere is an active region where plant roots engage with soil microorganisms. Root exudates affect microbial activity, augmenting nutrient cycling, promoting plant growth, increasing enzyme production, and benefiting soil health. Beneficial microbes enhance nutrient absorption, improve the structure of the soil, and safeguard plants against diseases; hence, they play a vital role in the stability of agroecosystems.

**Figure 2 ijms-26-00109-f002:**
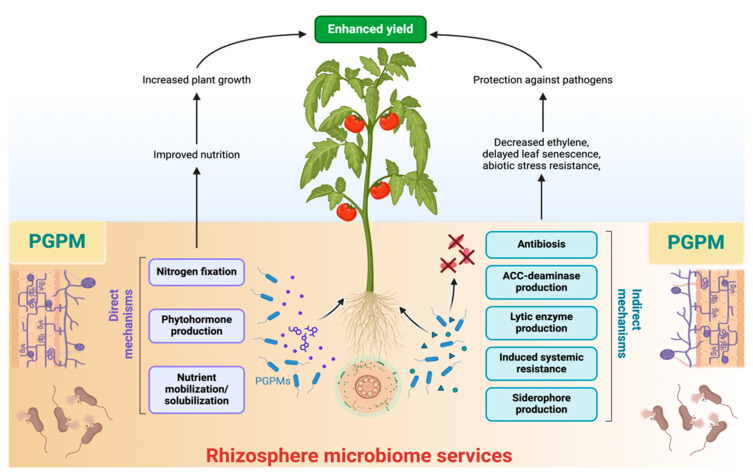
Soils in agroecosystems serve as a primary nutrient reservoir for crops, supplying necessary nutrients and water while also functioning as the most significant microbial reproductive bank for the rhizosphere microbiome. The plant offers many microhabitats, including both the endosphere and rhizosphere, for the development of the rhizosphere and plant microbiome. Beneficial plant–microbe interactions facilitate the rhizosphere microbiome’s services. Microbiome services improve plant growth, stress tolerance, and disease resistance, as well as host resilience to abiotic challenges such as cold, salinity, and drought, resulting in biomass production increasing up to fourfold and enhancing rhizosphere processes through their direct and indirect mechanisms. We created the conceptual diagram to provide an overview of rhizosphere microbiome services.

**Figure 3 ijms-26-00109-f003:**
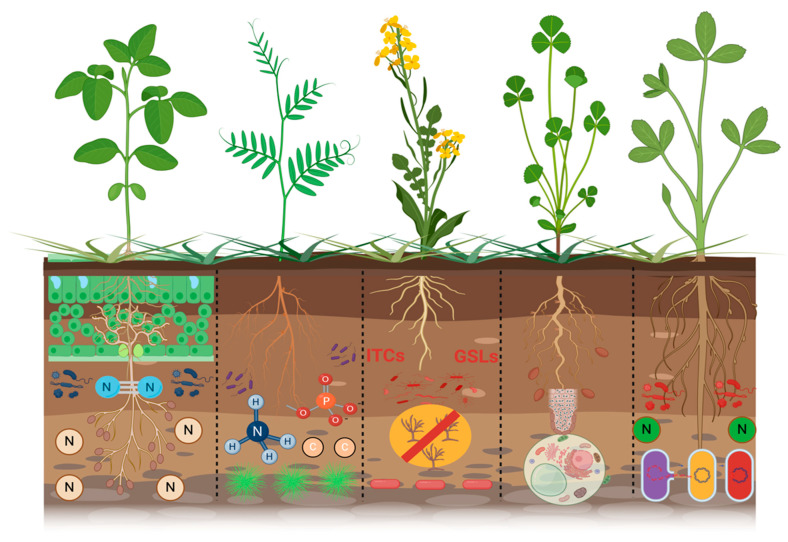
Various green manures serve to improve soil health. The diagram illustrates the roles of leguminous and non-leguminous green manures in agroecosystems. Leguminous green manures, including clover, Hairy Vetch, and peas, improve soil health chiefly by facilitating biological nitrogen fixation; hence, they diminish the reliance on synthetic fertilizers and augment nitrogen availability for succeeding crops. Non-leguminous green manures, including mustard, rye, and radish, enhance organic matter content, improve soil structure, and increase carbon sequestration. Both varieties of green manures enhance soil biodiversity by promoting microbial activity and nurturing beneficial microbial populations, such as nitrogen-fixing bacteria and mycorrhizal fungi. Furthermore, they recycle nutrients, inhibit soil-borne diseases, and enhance nutrient availability, promoting a nutrient-dense and biologically active soil ecosystem. The cumulative impacts lead to increased soil fertility, greater resilience against erosion and climate stress, and extended productivity of agroecosystems.

**Figure 4 ijms-26-00109-f004:**
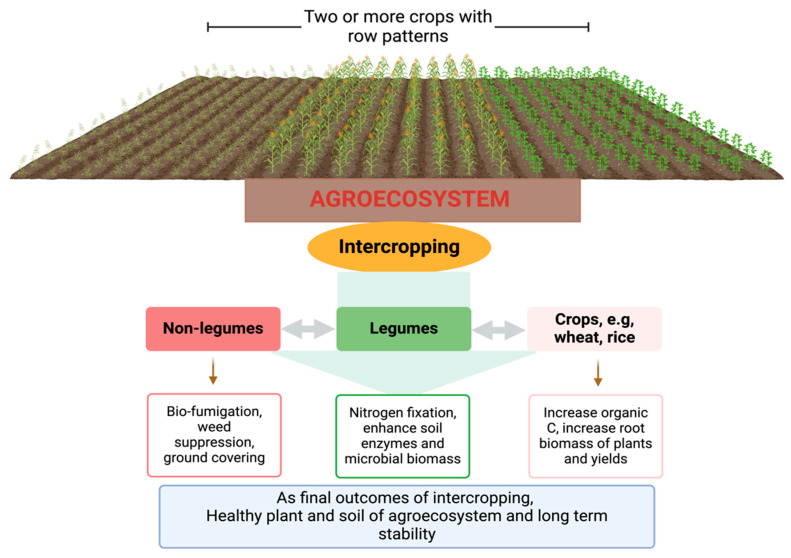
An overview of intercropping systems highlights their beneficial impacts on cash crops and long-term agroecosystem sustainability. Intercropping improves nutrient use efficiency, enhances soil health, promotes biodiversity, reduces pest and disease incidence, and increases crop resilience, contributing to stable yields and sustainable agricultural systems.

**Figure 5 ijms-26-00109-f005:**
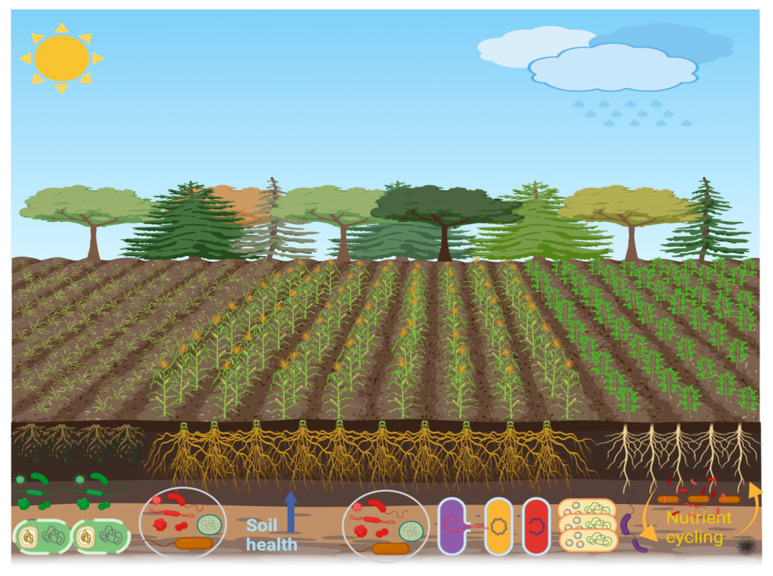
We present a conceptual depiction of the integration of crop rotation systems, tracing their journey from challenges to outcomes. Crop rotation systems that incorporate cash and legume crops can sustain agricultural yields, increase farmers’ income, and minimize greenhouse gas emissions through legumes’ biological nitrogen fixation, which partially replaces synthetic nitrogen inputs. Incorporating legumes into crop rotations can improve soil health by promoting microbial and enzyme activity, optimizing nutrient cycling, and enhancing carbon sequestration.

**Table 1 ijms-26-00109-t001:** Potential role and activities of PGPM in the rhizosphere for sustainable agriculture.

PGPM	Function and Role	References
*Penicillium* spp.	Promotion of plant growth, increase systemic resistance, and confrontation to *Pseudomonas syringae* pv. Tomato	[60,61]
*Trichoderma* spp.	Plant growth promotion, production of indole-3-acetic acid IAA, production of siderophores, resistance against soil-borne disease	[33,62]
*Aspergillus* spp.	Induces systemic resistance against necrotrophic fungus, plant growth promotion and phytohormones production	[63]
*Purpureocillium* spp.	Plant growth promotion and siderophores and enzyme production	[64]
*Talaromyces* spp.	Promotion of plant growth, phosphate solubilization, siderophores production	[65]
*Bacillus* spp.	Promotion of plant growth, Indole-3-acetic acid (IAA) and phosphate solubilization, siderophore production	[52]
*Azotobacter* spp.	Ammonia production, Indole-3-acetic acid, growth enhancement of canola and lettuce plant, Phosphorus solubilization	[23,66]
*Pseudomonas* spp.	Growth improvement of canola and tomato plants, hormones production such as IAA, production of siderophore	[67,68]
*Azospirillum* spp.	Indole-3-acetic acid (IAA) production, siderophore and exo-polysaccharides, and promote canola plant growth	[67,68]
*Streptomyces* spp.	*Streptomyces* inhibit *Curvularia oryzae* and volatile organic compounds (VOCs) releasing, promote plant growth as well as activities of defense-related enzymes	[69]

**Table 2 ijms-26-00109-t002:** Green manure and its beneficial effects.

Green Manure	Their Beneficial Effects	References
Oats, Hairy Vetch	Nutrient availability, soil organic matter, enhancing of soil enzymes and fungal biomass and soil inorganic N and P, reducing soil erosion	[34,86,102]
Winter Rye, Brassica	Biofumigation, reduction in soil-borne diseases, enhancement of soil enzymes and nitrogen fixation, as well as improvement in soil biodiversity and structure	[86,87,102]
Mustard, Barley	Soil and ground covering, nutrient cycling, and increasing microbial biomass, as well as weed suppression, Increasing fungi and bacteria ratio in soil	[20,87]
Red clover, Peas, Oats	Enhancement of biological activity in rhizosphere, nitrogen fixation and reduction of soil erosion, improve biological health, increase fungal abundance and soil enzyme activity	[103,104]
Cereal rye, Annual Ryegrass	Enhancement of SOM, control weed growth, control N_2_O emissions, NO^3−^ reduction, C sequestration, and sustain cash crop yields	[89,104,105,106]
Sunn hemp, Alfalfa, Cowpea	Reduce erosion of soil, nitrogen fixation, increase SOM, improve soil biodiversity, and weeds suppress	[105,107]
Oilseed Radish, Winter Wheat, Rapeseed	Reduce soil erosion, SOM improvement, loosen topsoil, enhance soil biodiversity, and suppress pest	[102,105,107]

## Data Availability

No data was used for the research described in the article.

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
