# Peer review of "Enhancing the Resilience of Agroecosystems Through Improved Rhizosphere Processes: A Strategic Review"

_ijms, 2024, doi:10.3390/ijms26010109_

Round 1

Reviewer 1 Report

Comments and Suggestions for Authors

The present review manuscript titled Enhancing the Resilience of Agroecosystems through Improved Rhizosphere Processes: A Strategic Review is expressing the challenges and problems in the agricultural field. This topic focuses on how optimizing rhizosphere processes can enhance the resilience of agroecosystems. The rhizosphere, a critical zone of interaction between plant roots and soil microbes, plays a vital role in nutrient cycling, soil structure maintenance, and plant health. This strategic review examines innovative approaches, such as the use of beneficial microbes, organic amendments, and intercropping, to improve rhizosphere functionality. The review also highlights the importance of these processes in mitigating abiotic and biotic stresses, thereby supporting sustainable agriculture. Enhancing rhizosphere processes is crucial for creating resilient agroecosystems capable of withstanding climate change, soil degradation, and resource scarcity. By improving nutrient efficiency, reducing reliance on synthetic inputs, and promoting biodiversity, these strategies contribute to sustainable crop production and environmental health. This research is important for guiding future agricultural practices, ensuring food security, and maintaining ecosystem services in the face of global environmental challenges.

General comments

The authors have constructed a highly informative and compelling narrative on agroecosystems, demonstrating significant depth and clarity. However, before final publication and acceptance, I would like to suggest several points that require further attention to ensure the manuscript reaches its full potential and I will recommend minor revisions before acceptance.

Specific comments

1. In the abstract, please add one or two sentences about agroecosystems to provide better context.

2. In the keywords, replace 'agroecosystem' with 'sustainable agroecosystem' for better specificity.  

3. Please add more recent references, particularly those related to agroecosystems, to strengthen the introduction.  

4. Kindly remove any repeated sentences from the introduction and other parts of the manuscript.

5. Please add a section on how green manure and rhizosphere microbes can contribute to rebuilding and improving degraded soil.

6. Why did the authors specifically choose the improvement of agroecosystems through these techniques as the focus of their study?

7. Add more information in Table 2.

8. Please include additional information along with the latest references in Section 7: Integrated Agricultural Practices for Improving Rhizospheric Processes."

Best wishes

Author Response

Dear Respected Reviewer,

Thank you very much for your valuable comments and suggestions.

We have revised our manuscript as follows.

1st Reviewer

The authors have constructed a highly informative and compelling narrative on agroecosystems, demonstrating significant depth and clarity. However, before final publication and acceptance, I would like to suggest several points that require further attention to ensure the manuscript reaches its full potential and I will recommend minor revisions before acceptance.

Answer: We sincerely appreciate the reviewer’s positive feedback and kind words regarding our manuscript. It is encouraging to hear that our work has been considered highly informative and compelling, with significant depth and clarity. We acknowledge the minor revisions suggested to further enhance the quality of the manuscript. We have carefully addressed each of the reviewer’s points and made the necessary changes to ensure the manuscript reaches its full potential. The revisions aim to strengthen the clarity, focus, and scientific rigor of the work. We are grateful for the opportunity to improve our manuscript further and appreciate the reviewer’s constructive suggestions and thoughtful review process. Thank you for your time and consideration.

Question:  In the abstract, please add one or two sentences about agroecosystems to provide better context.

Answer: We have added the sentence on agroecosystem. Line 19-20. Thank you

Question: In the keywords, replace 'agroecosystem' with 'sustainable agroecosystem' for better specificity. 

Answer: Replacement has been done. Line 41

Question: Please add more recent references, particularly those related to agroecosystems, to strengthen the introduction.

Answer: We have improved the introduction based on your valuable comments. Line 46-49, Thank you

Question: Kindly remove any repeated sentences from the introduction and other parts of the manuscript.

Answer: Modification has been done. Thank you

Question: Why did the authors specifically choose the improvement of agroecosystems through these techniques as the focus of their study?

Answer: Thank you for the technical question. Actually we opted to enhance agroecosystems through the techniques we discussed in our study, specifically rhizosphere management, green manure addition, and intercropping, as these methods have demonstrated increased resilience and sustainability in agricultural systems. Agroecosystems are intrinsically intricate, encompassing interactions among flora, microbes, soil, and the surrounding environment. The health and operation of these systems are essential for food security, environmental sustainability, and climate resilience. By emphasizing these strategies, we seek to underscore their significance for fostering soil health, augmenting nutrient cycling, enhancing microbial biodiversity, and diminishing dependence on synthetic inputs—all of which are crucial for the enduring stability of agroecosystems.  Moreover, as global issues like climate change and resource depletion escalate, enhancing agroecosystem processes through sustainable methods is increasingly crucial. These strategies enhance ecological equilibrium in agricultural systems while offering economic and environmental advantages, positioning them as crucial to the advancement of sustainable agriculture.

Question: Add more information in Table 2.

Answer: Table 2 has been updated as per the respected reviewer comments. Thank you

Question: Please include additional information along with the latest references in Section 7: Integrated Agricultural Practices for Improving Rhizospheric Processes."

Answer: We have improved the section based on your and other respected reviewer suggestions. Thank you Line 780, 810-824.

Reviewer 2 Report

Comments and Suggestions for Authors

Dear Authors, 

The manuscript entitled “Enhancing the Resilience of Agroecosystems through Improved

Rhizosphere Processes: A Strategic Review” is of interest for the readers and fits to the scope of this journal. The manuscript is well structured and written with a red thread; however, it must be written more focused as many part of the manuscript are repeated; Line 254-255 has been stated two times before within the manuscript.  E.g. “plant growth promoting microbes (PGPM)” must be written out first time, afterwards abbreviation can be used (this is not consistent throughout the manuscript). 

All paragraphs must be more related to the title of the manuscript. 

It seems like the paragraphs are written by different authors and the manuscript was not curated by one author before submission. Please curate the whole manuscript to avoid writing the same several times in other words and to unify the citations/references.

The figures of the manuscript are not necessary to understand the text, as they did not support the text. They are more less than a geographical abstract and some of the captions did not explain the figure. They would nicely fit in a textbook but do not provide any scientific information related to this review paper. They should be deleted from the manuscript. 

The abstract: tree-fourth of the abstract gives a short introduction of the background and state of the art. Only one-fourth deals with aim, findings and results of the current work. This part must be rewritten more focusing on “abstract of the manuscript” and shortened. 

Introduction: this part provides a good overview and background of the topic and is written straight forward. References are up to date. The last two sentences of this part give a summary of the findings of this review and must be moved to the abstract. 

The rhizosphere: its mechanism and roles: this part is dived into the rhizosphere processes and - engineering parts. The authors describe the rhizosphere and its role, e.g. interactions between organisms, roots and soil in detail as well as interventions which improve the rhizospheric processes. Line 156: CO2: please write CO2.

Role of soil microbes in the rhizosphere processes: here influences on microbial communities in soil are explained in detail, however, this should be linked more to the resilience of the agroecosystems. E.g. Line 313 ff, please provide some information, how these changes in community structure have effects on the soil/agroecosystem. 

Role of green manures in agroecosystems:  

This part is written in detail; however, some parts lack the relation to soil health (line 440 – 470, here the authors report on plant disease and did not link this to soil health). 

Line 364: Kataoka et al. please revise

Line 373: Chavarria et al., please revise

Line 389: Sun et al., please revise 

Line 387-391: who are these two sentences related to each other? Please describe in detail, as this is an important point. 

Line 392-393: what is meant by “during this phase”? what does it refer to? 

Table 2 must be strongly revised: Column “References” and column “their beneficial effects”:

E. g. references must be written like “Toom et al. “. Soil organic matter, nutrient availability are no beneficial effects. Enhanced availability e.g. could be a beneficial effect.

4.2. please define what is meant by “soil health” in this context. Otherwise, it is not clear why it has to be improved. 

e.g. Line 440, 449. “et al.” should not be written in italic. Please check this whole paragraph and revise. 

Line 481: delete “space” before “Therefore”.

Figure 3: Functions are not explained here. Please provide a detail caption of this figure, which explains this figure. 

Role of cropping systems in agroecosystems: Line 491-496 this is an overall introduction which should be moved to the beginning “introduction” of manuscript. 

Indicators for assessing soil health in agroecosystems: this paragraph needs to be moved further forward to better understand paragraph 3-5. It explains essential provides basic background information which are important for the whole text. 

Line 676: please delete “space” before “Additionally”.

Line 683-885: why are the aims of the study presented within this paragraph? 

Integrated agricultural practices for improving rhizospheric processes: this paragraph comes across as a sequence of sentences lacking any relation to one another. It must be improved. 

Concluding remarks and future directions: should be shortened a bit and may contain a “take home message”. The first paragraph of this part is no conclusion, it is more like an introduction. Please rewrite and shorten. Only line 788-798 match a summary. 

Further, line spacing must be consistent. 

Author Response

Dear Respected Reviewer,

Thank you very much for your valuable comments and suggestions.

We have revised our manuscript as follows.

Dear Authors, 

The manuscript entitled “Enhancing the Resilience of Agroecosystems through Improved Rhizosphere Processes: A Strategic Review” is of interest for the readers and fits to the scope of this journal. The manuscript is well structured and written with a red thread; however, it must be written more focused as many part of the manuscript are repeated; Line 254-255 has been stated two times before within the manuscript.  E.g. “plant growth promoting microbes (PGPM)” must be written out first time, afterwards abbreviation can be used (this is not consistent throughout the manuscript). 

Answer: We would like to thank the respected reviewer for their constructive comments and valuable suggestions, which have greatly contributed to improving the focus and quality of our manuscript entitled “Enhancing the Resilience of Agroecosystems through Improved Rhizosphere Processes: A Strategic Review.” Repetitions (e.g., Line 254–255): We have carefully reviewed the manuscript and removed repetitive content, ensuring that each concept or point is stated clearly and only once to maintain conciseness and avoid redundancy. Abbreviations (e.g., “plant growth-promoting microbes (PGPM)”): We have ensured that all abbreviations are consistently written out in full upon their first appearance, followed by their respective abbreviations. This has been thoroughly checked throughout the manuscript for consistency. We are grateful for your valuable feedback and appreciate the opportunity to revise and improve our manuscript. Your comments have been instrumental in enhancing the focus, clarity, and overall quality of the work. Thank you for your time and effort.

Question: All paragraphs must be more related to the title of the manuscript. 

It seems like the paragraphs are written by different authors and the manuscript was not curated by one author before submission. Please curate the whole manuscript to avoid writing the same several times in other words and to unify the citations/references.

The figures of the manuscript are not necessary to understand the text, as they did not support the text. They are more less than a geographical abstract and some of the captions did not explain the figure. They would nicely fit in a textbook but do not provide any scientific information related to this review paper. They should be deleted from the manuscript.

Answer: We sincerely thank the respected reviewer for their constructive feedback and valuable suggestions to improve the quality of our manuscript. Below, we address the specific comments provided:

  1. Figure Captions and Relevance:
    We have carefully reviewed all the figures and their captions. Based on the reviewer's comments, we have made significant improvements to the figure captions to ensure they clearly explain and support the content presented in the text. Furthermore, figures that do not add scientific value or directly relate to the core focus of the review have been removed. This ensures that all remaining figures contribute meaningfully to the manuscript's narrative and scientific rigor.
  2. Consistency of Content and Citations:
    We acknowledge the reviewer’s concern regarding inconsistencies in writing style and redundancy across paragraphs. To address this, we have thoroughly curated the manuscript to unify its tone, ensure a logical flow of ideas, and remove any repetitive content. Additionally, the manuscript has been carefully reviewed to standardize all citations and references in accordance with the journal’s guidelines.
  3. Relation to the Title:
    All sections and paragraphs have been revised to ensure they are directly aligned with the manuscript's title, “Enhancing the Resilience of Agroecosystems through Improved Rhizosphere Processes: A Strategic Review.” The content now maintains a strong focus on the central theme of the review, emphasizing rhizosphere processes and their role in enhancing agroecosystem resilience.

We are grateful for the reviewer’s insightful comments, which have greatly enhanced the scientific quality, clarity, and focus of the manuscript. Thank you for your time and valuable suggestions.

Question: The abstract: tree-fourth of the abstract gives a short introduction of the background and state of the art. Only one-fourth deals with aim, findings and results of the current work. This part must be rewritten more focusing on “abstract of the manuscript” and shortened. 

Response: Dear respected reviewer, Thank you very much for your valuable feedback and constructive suggestions. We have carefully revised the abstract to address your concerns. Specifically, we have shortened the background and state-of-the-art portion and provided a clearer focus on the aims, findings, and results of the current work. Additionally, we have added a concluding sentence, as you suggested, enhancing the clarity and relevance of the abstract. We sincerely appreciate your thoughtful input, which has helped improve the quality of the manuscript. Please let us know if there are any further suggestions or adjustments needed. Thank you for your time and guidance.

Question: Introduction: this part provides a good overview and background of the topic and is written straight forward. References are up to date. The last two sentences of this part give a summary of the findings of this review and must be moved to the abstract. 

Response: we have added a concluding sentence, as you suggested, enhancing the clarity and relevance of the abstract.

Question: The rhizosphere: its mechanism and roles: this part is dived into the rhizosphere processes and - engineering parts. The authors describe the rhizosphere and its role, e.g. interactions between organisms, roots and soil in detail as well as interventions which improve the rhizospheric processes. Line 156: CO2: please write CO2.

Response: Modification has been done, Line 162

Question: Role of soil microbes in the rhizosphere processes: here influences on microbial communities in soil are explained in detail, however, this should be linked more to the resilience of the agroecosystems. E.g. Line 313 ff, please provide some information, how these changes in community structure have effects on the soil/agroecosystem. 

Response: Respected reviewer, we have made modifications and added more reference-based information to lines 334-340. Thank you

Question: Role of green manures in agroecosystems: This part is written in detail; however, some parts lack the relation to soil health (line 440 – 470, here the authors report on plant disease and did not link this to soil health). 

Answer: Respected reviewer, we have made modifications as per your valuable question, and added more reference-based information to lines 513-530. Thank you

Question: Line 364: Kataoka et al. please revise, Line 373: Chavarria et al., please revise, Line 389: Sun et al., please revise.

Answer: References has been updated. Line 380, 389. Thank you

Question: Line 387-391: who are these two sentences related to each other? Please describe in detail, as this is an important point. 

Answer: Respected reviewer, we have made modifications as per your valuable question, and added more reference-based information to lines 404-411. Thank you

Question: Line 392-393: what is meant by “during this phase”? what does it refer to? 

Answer: The information was invalid; thank you for pointing it out, we have removed the sentence.

Question: Table 2 must be strongly revised: Column “References” and column “their beneficial effects”:

Answer: References and table 2 has been updated as per the reviewer comments. Thank you

Question: 4.2. please define what is meant by “soil health” in this context. Otherwise, it is not clear why it has to be improved. 

Answer: We have added the possible information regarding soil health. Line 428-435.

Question: e.g. Line 440, 449. “et al.” should not be written in italic. Please check this whole paragraph and revise.

Answer: Modification has been done throughout the manuscript. Thank you

Question: Line 481: delete “space” before “Therefore”.

Answer: Space removed. Line 530.

Question: Figure 3: Functions are not explained here. Please provide a detail caption of this figure, which explains this figure. 

Answer: We have updated the caption of Figure 3 to reflect the various functions of green manure. Line 535-547. Thank you.

Question: Role of cropping systems in agroecosystems: Line 491-496 this is an overall introduction which should be moved to the beginning “introduction” of manuscript. 

Answer: We've moved the paragraph to the beginning of the introduction. Thank you.

Question: Indicators for assessing soil health in agroecosystems: this paragraph needs to be moved further forward to better understand paragraph 3-5. It explains essential provides basic background information which are important for the whole text. 

Answer: Thank you for your valuable suggestion, and we have moved the paragraph Line 722-725.

Question: Line 676: please delete “space” before “Additionally”.

Answer: Space deleted. Line 736

Question: Line 683-885: why are the aims of the study presented within this paragraph? 

Answer: We replaced the word paragraph instead of review. Line 743

Question: Integrated agricultural practices for improving rhizospheric processes: this paragraph comes across as a sequence of sentences lacking any relation to one another. It must be improved. 

Answer: We have rearranged the sentences and added a few new words. Line 777-778, 807-821. Thank you.

Question: Concluding remarks and future directions: should be shortened a bit and may contain a “take home message”. The first paragraph of this part is no conclusion, it is more like an introduction. Please rewrite and shorten. Only line 788-798 match a summary. Further, line spacing must be consistent. 

Answer: We have shortened the conclusion paragraph and rearranged the line spacing.

Round 2

Reviewer 2 Report

Comments and Suggestions for Authors

accept in present form